# Influence of Natural Mordenite Activation Mode on Its Efficiency as Support of Nickel Catalysts for Biodiesel Upgrading to Renewable Diesel

**DOI:** 10.3390/nano13101603

**Published:** 2023-05-10

**Authors:** Konstantina Fani, Sotiris Lycourghiotis, Kyriakos Bourikas, Eleana Kordouli

**Affiliations:** 1School of Science and Technology, Hellenic Open University, Parodos Aristotelous 18, 26335 Patras, Greece; fani.konstantina@ac.eap.gr (K.F.); sotirislyk@gmail.com (S.L.); bourikas@eap.gr (K.B.); 2Department of Chemistry, University of Patras, 26504 Patras, Greece

**Keywords:** biodiesel, renewable diesel, nickel mordenite catalysts, deposition–precipitation, flake-like nickel

## Abstract

In the present work, natural mordenite originated from volcanic soils in Greek islands, activated using HCl solution and HCl solution followed by NaOH solution, was used as support for preparing two metallic nickel catalysts (30 wt.% Ni). The catalysts were thoroughly characterized (XRF, N_2_ adsorption–desorption, SEM, XRD, TEM, H_2_-TPR, NH_3_-TPD) and evaluated for biodiesel upgrading to green (renewable) diesel. Double activation of natural mordenite optimized its supporting characteristics, finally resulting in a supported nickel catalyst with (i) enhanced specific surface area (124 m^2^ g^−1^) and enhanced mean pore diameter (14 nm) facilitating mass transfer; (ii) easier nickel phase reduction; (iii) enhanced Ni^0^ dispersion and thus high active surface; (iv) balanced population of moderate and strong acid sites; (v) resistance to sintering; and (vi) low coke formation. Over the corresponding catalyst, the production of a liquid consisting of 94 wt.% renewable diesel was achieved, after 9 h of reaction at 350 °C and 40 bar H_2_ pressure, in a semi-batch reactor under solvent-free conditions.

## 1. Introduction

The alarming rate of climate change in combination with the diminishing of petroleum reserves have spurred to the urgent need to develop sustainable alternative fuels suitable for replacing fossil fuels. Although biodiesel (FAMEs) produced using transesterification of triglyceride biomass with a light alcohol [1,2] has already entered in the fuel market as a very promising alternative to petro–diesel, its utilization without mixing with the latter is still problematic. Biodiesel is very unstable (at low temperature it is prone gelling and at high temperature it grows mold), it may damage filters and pipes of the existing vehicles and has lower fuel efficiency than petro–diesel [2]. Considering these drawbacks, it is of great interest to design catalytic systems to upgrade biodiesel to the next generation, higher value fuel consisting of hydrocarbons in the diesel range, so-called renewable or green diesel.

Hydrotreatment has proven to be a very promising process for upgrading FAMEs to renewable diesel [3,4,5]. This process aims to their selective deoxygenation (SDO), avoiding extended cracking to hydrocarbons with carbon atoms much less than those existing in the initial fatty acids. Several catalytic systems have been studied for SDO. Noble metal catalysts have proven very effective [6,7,8] but too expensive for industrial applications. Sulfided NiMo [9] and CoMo [10,11] catalysts, which are traditionally used for oil refining, have also been studied. However, the need for continuous sulfidation upon SDO process makes their use undesired. In the last decades, base metals have attracted researchers’ interest for the development of SDO monometallic, bimetallic or multi-metallic catalysts using various supports (Al_2_O_3_, SiO_2_, ZrO_2_, Zeolites, etc.) [12,13]. The acidity of the supports contributes to the creation of bifunctional catalysts. Bifunctionality of the catalysts is crucial for the performance of SDO catalysts [14,15]. The exploitation of abundant minerals as catalytic supports is of great importance from both economic and environmental points of view. Natural mordenite could be a very promising mineral, as it is characterized by rather large micropores (~7 Å), enhanced thermal stability up to 900 °C and a texture that makes it suitable as a catalyst [16,17,18] or a catalytic support [5,19,20,21]. The main channels of mordenite are connected by tortuous pores that are too small, which are not accessible for the most organic molecules, resulting in mass transport limitations and rapid deactivation. Moreover, due to their microporous structure, the coke formed only covers active sites but also blocks the pores. Thus, the major drawback of mordenite-based catalysts is the narrow and non-interconnected channels and cavities. This imposes diffusional limitations on the reactions, affecting catalytic performance [22].

Liu et al. [23] showed that mesoporous supports (2 nm < pore diameter < 50 nm) facilitating the diffusion of reactants are ideal for vegetable oil SDO processes. Supports with mesoporous structure and relatively high specific surface area are considered the most suitable for metallic nickel catalysts. In contrast, the pores of microporous supports (pore diameter < 2 nm) are easily blocked by nickel-supported species, thus diminishing the specific surface area of the final catalyst [24].

Previous studies have shown that acid treatment of mordenite improves its textural characteristics, increasing the specific surface area without a significant change in its pore structure, resulting in microporous material [5,16,17,25]. In contrast, alkali treatment of mordenite increases its mesoporosity creating inter- and intracrystalline pores [26,27].

The reaction network of biodiesel SDO upon Ni-supported catalysts (Appendix A) begins with the saturation of C=C bonds. This is followed by O–C bond hydrogenolysis and the reduction in free fatty acids (FFAs) to aldehydes and then to alcohols. The aldehydes are decarbonylated (DeCO) towards hydrocarbons having one carbon atom less than the corresponding aldehydes, while the alcohols are hydrodeoxygenated (HDO) to hydrocarbons having the same carbon atoms. In parallel, FFAs can be decarboxylated (DeCO_2_), producing hydrocarbons with one carbon atom less (minor extent) or can react with produced alcohols towards high molecular weight esters. The latter, in a next slow step, are hydrogenolyzed, producing more hydrocarbons. Details of the corresponding reaction mechanism have been provided elsewhere [5].

Nickel catalysts supported on acid-treated natural mordenite have proven quite promising for biodiesel transformation to green diesel [5]. In the present work, we aim to exploit these findings, targeting more efficient Ni/mordenite catalysts. This goal is pursued by changing the texture of microporous mordenite to a more mesoporous one. Thus, we are studying the influence of the activation mode of natural mordenite (acid and acid–base treatment) in order to improve both its texture and its supporting properties. In addition, we examined the effect of reaction temperature on biodiesel upgrading to renewable diesel using the improved catalyst.

## 2. Materials and Methods

### 2.1. Materials

Natural mordenite was kindly provided by TECHNOTOPIA Ltd. (Athens, Greece). Hydrochloric acid solution (Carlo Erba, Emmendingen, Germany, 35% *w*/*w*), ammonium nitrate (Merck, Darmstadt, Germany) and sodium hydroxide (Lach-Ner, Neratovice, Czech Republic) were used for mordenite treatment. Nickel (II) nitrate hexahydrate (Ni(NO_3_)_2_‧6H_2_O) (Alfa Aesar, Ward Hill, MA, USA) and urea (CO(NH_2_)_2_) (Duchefa, Haarlem, The Netherlands) were used for catalysts synthesis. Biodiesel obtained from a local biodiesel company P. N. Pettas S.A., Edible Oils and Fats-Biofuel-Soap Manufacturers (Patras, Greece).

### 2.2. Acid and Acid–Base Treatment of Natural Mordenite

Acid activation of natural mordenite was performed as described previously [5] using treatment with hydrochloric acid aqueous solution (2 M, mass of solid to solution volume ratio: 1 g/20 mL) followed by drying (110 °C overnight), calcination (500 °C for 2 h) and grinding. The support obtained is symbolized as MO_A_.

Acid–base treatment of natural mordenite was performed by treating a MO_A_ sample with NaOH aqueous solution (2 M) in a solid mass-to-solution volume ratio equal to 1 g/15 mL. The suspension was stirred at 65 °C for 30 min, filtered, washed several times with distilled water and dried at 110 °C overnight. The solid obtained and the NH_4_NO_3_ aqueous solution (1 M) were mixed in a round-bottomed flask, retained at 80 °C for 24 h under stirring, followed by cooling at 25 °C. The suspension was filtered and washed several times with distilled water until the pH of the filtrate reached 7. The obtained solid was dried at 110 °C for 12 h, calcined at 500 °C for 2 h and grinded to become fine grained. The support obtained is symbolized as MO_AB_.

### 2.3. Nickel Catalysts Preparation

The deposition–precipitation method was used for the catalysts’ preparation with 30 wt.% Ni. They are symbolized as NiMO_x_, where x represents the activation mode of natural mordenite (A or AB). Details of the preparation procedure are provided in Appendix A.

### 2.4. Catalysts’ Characterization

The supports and catalysts’ compositions were determined using X-ray Fluorescence spectroscopy (XRF). Textural characteristics of the samples were determined using nitrogen physisorption. X-ray powder diffraction (XRD) was used to identify crystal phases of the samples. The morphology of the samples was observed using scanning electron microscopy (SEM). Metallic nickel crystal size distribution was calculated using transmittance electron microscopy (TEM) results. Samples’ acidity was determined using temperature-programmed desorption of ammonia (NH_3_-TPD). Reducibility of the catalyst precursors was studied using temperature-programmed reduction with H_2_ (H_2_-TPR). Combustion elemental analysis was used to determine the coke deposited on the spent catalyst samples. Experimental details and setups used are provided in Appendix A.

### 2.5. Catalysts’ Evaluation

Catalyst performance was evaluated in a high-pressure (40 bar) semi-batch reactor fed with H_2_ (100 mL/min) at reaction temperatures 310, 330 and 350 °C under solvent-free conditions [28]. For more details, see Appendix A.

## 3. Results and Discussion

### 3.1. Catalysts’ Characterization

In the present work, nickel catalysts supported on (singly-acid and doubly-acid/base) activated natural mordenite were prepared using the deposition–precipitation method. Ni loading (30 wt.%) and Si/Al atomic ratio in the natural mordenite as well as the activated supports (Table 1) were determined using XRF.

#### 3.1.1. Textural Properties

The porosity and the surface area of the supports and catalysts were determined using the nitrogen adsorption–desorption method. The isotherms obtained are shown in Appendix A. The MO_nat._ isotherm, according to IUPAC classification, is of type I; the activated supports and the corresponding nickel catalysts isotherms are of type IV. They present a steep curvature at low relative pressure, which is characteristic of microporous materials (pore diameters ≤ 2.0 nm) and a hysteresis loop at higher relative pressures indicating the development of a mesoporous network of pores (2.0 ≤ pore diameters ≤ 50.0 nm). The activated supports present as an H1 hysteresis loop, which is characteristic of porous materials exhibiting well-defined cylindrical-like pores. The NiMO_A_ and NiMO_AB_ catalysts show an H4-type hysteresis loop, indicating the formation of narrow slit pores [29].

The textural characteristics of the materials are illustrated in Table 1. The acid treatment of MO_nat._ seems to drastically enhance the S_BET_, mainly the surface of the micropores. This is due to the removal of Na^+^, K^+^, Mg^2+^, Ca^2+^ and Fe^3+^ from the zeolite channels. Upon acid treatment of MO_nat._, Al^3+^ ions are removed from its framework (dealumination process), resulting in an increase in Si/Al ratio. This process leads to the hydrolyzation of Si–O–Al bonds, producing –Si–OH and –Al–OH, and forming vacant sites in the zeolite framework. Further treatment of this material with a base leads to Si/Al ratio diminution of MO_A_ (desilication process) and to a further enhancement of S_BET_. Acid and acid–alkaline treatment of MO_nat._ also lead to an increase in both pore volume and pore diameter (Table 1) [30].

The addition of Ni to the activated supports seems to decrease the corresponding specific surface area, mainly by blocking the micropores. This is reflected by the substantial decrease in S_micro_ values (Table 1). On the other hand, the values of the specific pore volume are higher on the nickel catalysts than those of the corresponding supports. This indicates that new pores are created after nickel phase deposition. This becomes more obvious in Figure 1, which presents the pore size distribution curves of the materials. Indeed, the NiMO_A_ and NiMO_AB_ catalysts’ curves appear maximal in the mesoporous range 20–30 nm. This is a positive fact, reducing mass transfer limitations upon reactions over such catalysts.

#### 3.1.2. Morphology of the Materials

Figure 2 presents SEM images of support materials (MOA, MOAB) and nickel catalysts (NiMOA, NiMOAB). Inspection of this figure shows that MOA exhibits a plate-like morphology (Figure 2a), which seems to be created by the adhesion of fibrous structures. On the other hand, acid–base treatment made the fibrous structure in MOAB more visible (Figure 2b). Deposition of the nickel phase on the above supports created flower-like structures that covered the corresponding supports (Figure 2c,d). Combining the new structures with the N2 physisorption results discussed in Section 3.1.1, one can conclude that these are responsible for the creation of new mesopores.

#### 3.1.3. Structural Properties of the Materials

Figure 3 presents the XRD patterns of the activated mordenite supports (MO_A_ and MO_AB_) and the corresponding catalysts (NiMO_A_ and NiMO_AB_). Inspection of this figure shows that zeolite framework remains almost intact after either acid or acid–alkali treatment of the parent material or after the deposition of the nickel phase. However, a slight shift towards low 2 theta angles of mordenite diffraction peaks was observed after acid–alkali treatment. This could be attributed to the desilication mentioned above and is in accordance with the increase in S_BET_ value (Table 1). The diffraction peaks at 2θ 44.58, 51.80 and 76.31° are attributed to the metallic nickel crystals (Ni^0^) with (111), (200) and (220) planes (PDF 87-0712). The diffraction peaks at 2θ 37.23, 43.29 and 62.92° are attributed to the NiO (111), (200) and (220) planes, respectively (PDF 65-6920). Metallic nickel is the main crystal phase detected in the NiMO_A_ and NiMO_AB_ catalysts. In addition, NiO diffraction peaks are rather obvious in the NiMO_A_ catalyst. This indicates that the NiO species formed upon thermal treatment (at 400 °C under Ar) before reduction interact more strongly with the MO_A_ than the MO_AB_ support. The mean crystal size values of the Ni^0^ (MCS_Ni_^0^) were estimated from the XRD data at 2θ equal to 51.8° when using Scherrer’s equation (Table 1). In both catalysts, these values were found to be almost the same and equal to 10 nm. This small value is evidence of very good dispersion of the nickel phase. However, deconvoluting the overlapped peaks corresponding to NiO (2θ: 43.29°) and Ni^0^ (2θ: 44.58°), we made a rough estimation of the (Ni^0^/NiO) ratio and found a value of about two in the NiMO_A_ catalyst, while this ratio became five in the NiMO_AB_ catalyst. This is in good agreement with H_2_-TPR results discussed in the next subsection.

Figure 4 shows representative TEM images of the catalysts, which prove that the nickel phase is evenly distributed on the supports’ surfaces. The nickel particle size distributions of the NiMO_A_ and NiMO_AB_ catalysts were also determined (Figure 4 right) using statistical analysis of about 250 particles. As can be seen, the mean particle size of Ni^0^ is ~7 nm for both catalysts, in good agreement with the XRD results (Table 1).

#### 3.1.4. Reducibility and Acidity Characteristics

In order to determine the suitable reduction temperature for the catalysts’ activation and to further investigate the strength of the metal−support interactions and thus reducibility of the NiO to Ni^0^, H_2_-TPR experiments were performed using catalyst precursor samples (after Ar treatment at 400 °C and before reduction).

Figure 5a shows the H_2_-TPR curves obtained. Inspection of the curve of the NiMO_A_ sample reveals that its NiO is reduced in a wide temperature range (235–542 °C). This means that well-dispersed NiO with various interaction strengths with the MO_A_ support was formed upon preparation [5]. A shoulder appearing in the range 542–625 °C could be attributed to the reduction in Ni^2+^—species incorporated in the support surface layers [31]. As for the NiMO_AB_ sample, the corresponding reduction curve shows two peaks. The first one, with a maximum at ~200 °C, indicates the existence of NiO weakly interacting with the MO_AB_ surface. The second and more intense peak, appearing at the 235–542 °C temperature range, indicates that the main part of NiO interacts moderately with the support surface. It should also be stressed that the NiMO_AB_ reduction curve does not present the aforementioned shoulder in the range 542–625 °C, indicating that incorporation of Ni^2+^ species in the support surface layers did not take place in this sample. The H_2_-TPR results confirm our previous conclusion, drawn using XRD analysis, that there is a stronger interaction between NiO species and the MO_A_ support. This is also in good agreement with the fact that the total amount of H_2_ consumed for the reduction in NiO supported on MO_AB_ is higher than that consumed for the NiO supported on MO_A_; however, the two samples have the same nickel loading.

NH_3_-TPD experiments were performed to investigate the acidity of the supports and the nickel catalysts [5,32]. Figure 5b illustrates the corresponding curves. Based on these curves, we calculated the total acid site populations and their distribution according to their strength (Table 2). All these curves were deconvoluted into four peaks. A low temperature desorption peak, with a maximum at ~106 °C, has been attributed to desorption of NH_3_ physisorbed on the sample’s surface. The corresponding amount was not taken into account for the calculation of the surface acid sites’ population. A second peak, appearing at ~190 °C, is attributed to weak acid sites. The third peak, with a maximum at ~300 °C, is attributed to surface acid sites with moderate acidity. The forth peak, at ~400 °C, is attributed to the strong acid sites. Table 2 involves the total acidity values as well as the percentages of acid sites according to their strength. Inspection of Figure 5b and Table 2 shows that the number of acid sites of MO_AB_ is higher than that of MO_A_. The acidity of the supports is mainly attributed to Brønsted acid sites corresponding to the hydroxyl group formed on the oxygen atom that bridges an aluminum atom with a silicon one, so a negative charge occurs, which can be recompensed by the proton (H^+^). Both supports exhibit enhanced population of weak acid sites (Table 2). The alkali treatment of MO_A_ leads to its desilication (Table 1) and to the enhancement of the SSA. As a result, the population of the aforementioned weak acid sites increases on this support. The addition of nickel decreases the weak acid sites indicating that probably the deposition precipitation mechanism of Ni^2+^ ions involves a first ion exchange step. On the other hand, nickel deposition creates a significant population of intermediate and strong acid sites (Figure 5b and Table 2). The latter could be due to the empty *d* orbitals of Ni^0^ (Lewis acid sites), which is better dispersed on the MO_A_ surface.

### 3.2. Catalysts’ Evaluation

The evaluation of the catalysts’ performance for biodiesel upgrading to renewable diesel was studied in a high-pressure semi-batch reactor without solvent, in a volume of biodiesel-to-catalyst mass ratio equal to 1 g/100 mL and hydrogen pressure of 40 bar. The GC analysis of the liquid product after 9 h reaction time is shown in Figure 6. Inspection of this figure shows that both catalysts (NiMO_A_ and NiMO_AB_) reached almost total conversion of the feed even at 310 °C. The double activation mode (acid–alkali) of natural mordenite seems to almost double the hydrocarbon production in the green diesel range from 27 wt.% to 52 wt.%. This improvement seems to take place at the expense of the intermediate high molecular weight esters (HMWE), the hydrodeoxygenation of which is considered to be the slowest reaction step under solvent-free conditions [33]. The improved catalytic performance of the NiMO_AB_ catalyst could be attributed to (i) its enhanced specific surface area and mean pore diameter (S_BET_ and d_BJH_ in Table 1) facilitating mass transfer; (ii) its easier reduction (Figure 5a) resulting in enhanced Ni^0^ dispersion (Figure 4) and thus high active surface; (iii) its balanced population of moderate and strong acid sites (Figure 5b and Table 2), which is expected to result in low coke formation (see below).

Based on the above findings concerning the influence of the mordenite activation mode on the performance of the final catalysts, we tested the most active one (NiMO_AB_) at higher reaction temperatures (330 and 350 °C) in order to further improve the yield of the process. The results presented in Figure 6 show that the intermediate products (acids and esters) are converted to final products to a greater extent (hydrocarbons in the diesel range) as the reaction temperature increases. Indeed, as high as 94 wt.% of the liquid products consisted of renewable diesel after 9 h of reaction at 350 °C.

It is well known that nickel catalysts used for SDO of triglycerides and relative compounds favor the DeCOx (DeCO and DeCO_2_) pathway instead of HDO (Appendix A), as hydrocarbons with odd carbon atoms are the main fraction of the hydrocarbons produced [24,34]. Figure 7 confirms that this is the case for all catalysts studied in the present work as well.

Figure 8 illustrates the kinetics of biodiesel hydrotreatment obtained over the NiMO_AB_ catalyst at 350 °C, taken as a typical example. This figure shows the variation in the biodiesel conversion (blue curve) and the yields of various liquid compounds produced in relation to time. Inspection of this figure shows that almost total conversion was reached after 4 h of reaction, whereas the yields of acids and esters passed through a maximum and then decreased, confirming that these are intermediate products. The yield of hydrocarbons increased monotonically with time, indicating that these are final products. Similar kinetics were obtained at all temperatures (310, 330 and 350 °C) over the NiMO_AB_ catalyst, and the results are presented in Appendix A. Based on these data, we attempted a kinetic analysis of the process.

Previously reported kinetic studies use the Langmuir–Hinshelwood or Eley–Rideal mechanism to describe the SDO process of FAME and triglyceride biomass for renewable diesel production. These models describe the experimental results excellently in several cases but are quite complex and require a lot of kinetic data in order for the reaction rate constants of the various reactions involved to be determined [7,35,36]. However, some recent studies [33,37,38,39], attempting to simplify the perplexing mechanism of triglyceride biomass SDO, adopt pseudo-first-order reaction steps. Following a similar approach, we considered that the transformation from biodiesel to green diesel obeys a pseudo-first-order reaction rate law. Based on the kinetic data obtained from the NiMO_AB_ catalyst for hydrocarbon production (Appendix A), and using Equation (1), we calculated the apparent reaction constant k at the three reaction temperatures (310, 330 and 350 °C).
−ln(1 − Y) = k × t,(1)
where Y is the hydrocarbon yield and t denotes reaction time.

Figure 9 shows that introduction of the calculated k values in an Arrhenius plot results in a very good linear correlation (R^2^ = 0.99999), confirming the assumption made for the reaction rate law. Based on the straight-line slope, we calculated the corresponding activation energy Eα = 64.4 ± 2.7 kJ mol^−1^. This value corresponds to the rate-determining steps, namely the SDO of FFAs and HMWE [33]. An Ea value higher than 40 kJ mol^−1^ also ensures that our results were obtained under kinetic regime, and the mass transfer phenomena’s influence on the activity results is negligible.

### 3.3. Spent Catalysts’ Characteristics

The characterization results of the spent catalysts are presented in Table 3. They show a dramatic decrease in S_BET_ in all cases in comparison with the values of the corresponding fresh catalysts (Table 1). XRD and TEM analysis of the spent catalysts (Table 3, Appendix A) show that sintering of nickel phase took place. This phenomenon could be one of the reasons for the aforementioned decrease in S_BET_. The sintering is more intense in the case of the NiMO_A_ catalyst, as according to TEM results, the mean size of the Ni^0^ nanoparticles increases 5.5 times (from 7.35 nm to 40.45 nm) and according to XRD results, the mean size of the Ni^0^ nanocrystals increases 1.7 times (from 11 nm to 19 nm). In contrast, the relevant changes observed in the case of the NiMO_AB_ catalyst correspond to an increase in the mean size of the Ni^0^ nanoparticles by 3.2 times (from 6.83 nm to 21.71 nm, TEM) and the mean size of the Ni^0^ nanocrystals by 1.5 times (from 10 nm to 15 nm, XRD). This effect is also combined with another negative effect, the coke deposited on catalysts’ surface. Coke formation is generally low in the studied catalysts. However, it is slightly higher in the case of the NiMO_A_ catalyst and seems to decrease with the reaction temperature in the case of the most active NiMO_AB_ catalyst (Table 3). This behavior has previously been mentioned several times for nickel catalysts [33,40].

## 4. Conclusions

The main conclusions drawn from the present study are summarized as follows:Natural mordenite is a promising support for Ni catalysts used for the SDO of biodiesel to green diesel;Double activation of natural mordenite using acid (HCl) solution followed by alkaline (NaOH) solution optimized its supporting characteristics, finally resulting in a supported nickel catalyst with (i) enhanced specific surface area and mean pore diameter facilitating mass transfer; (ii) easier nickel phase reduction (iii) enhanced Ni0 dispersion and thus high active surface; (iv) balanced population of moderate and strong acid sites; (v) resistance to sintering; and (vi) low coke formation.

## Figures and Tables

**Figure 1 nanomaterials-13-01603-f001:**
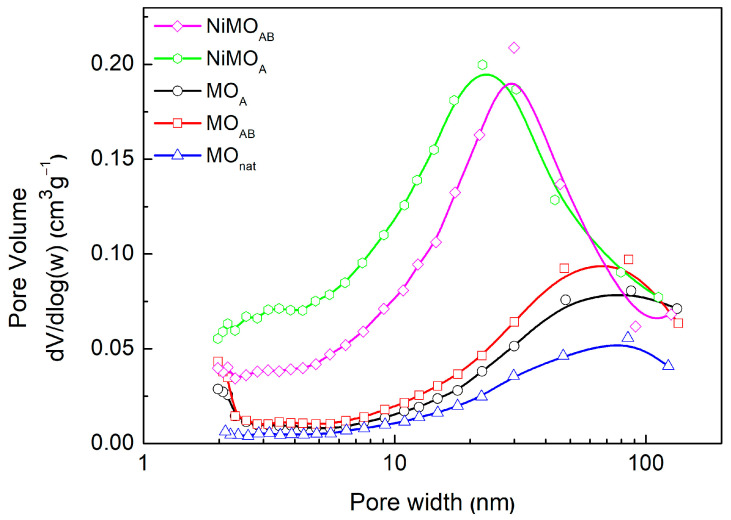
Pore size distributions of MO_nat._, MO_A_, MO_AB_, NiMO_A_ and NiMO_AB_.

**Figure 2 nanomaterials-13-01603-f002:**
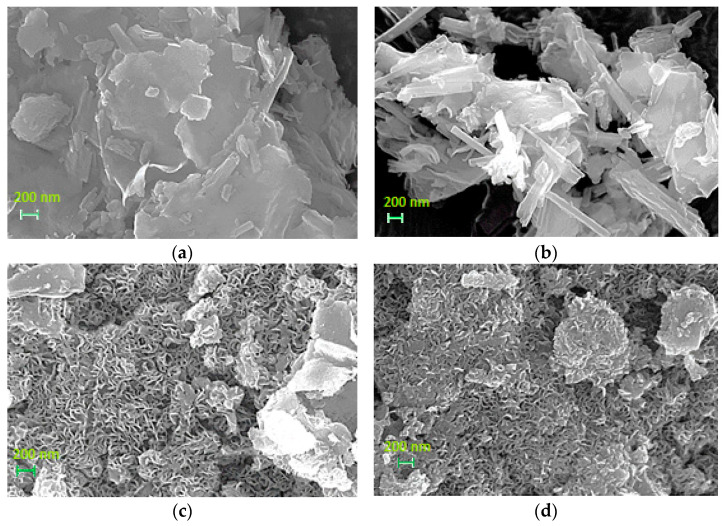
SEM images of (**a**) MO_A_, (**b**) MO_AB_, (**c**) NiMO_A_ and (**d**) NiMO_AB_, at magnification 25,000 KX.

**Figure 3 nanomaterials-13-01603-f003:**
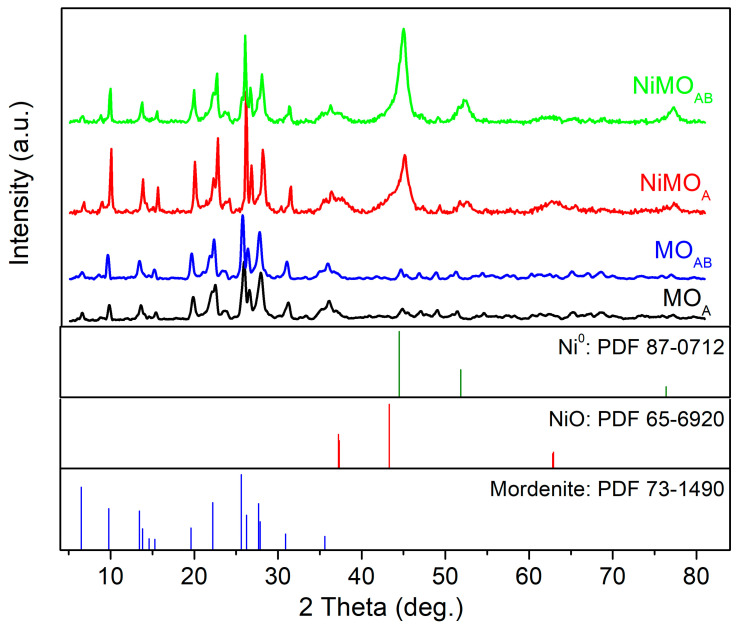
XRD patterns of MO_A_, MO_AB_, NiMO_A_ and NiMO_AB_.

**Figure 4 nanomaterials-13-01603-f004:**
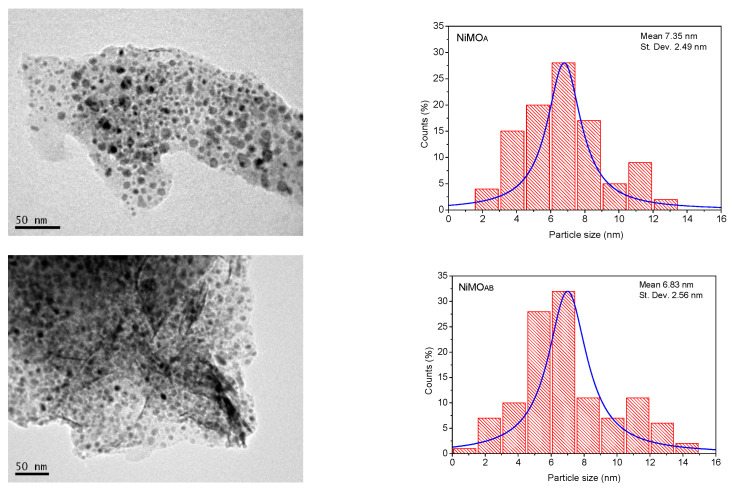
TEM images and nickel particle size distributions for NiMO_A_ and NiMO_AB_ catalysts.

**Figure 5 nanomaterials-13-01603-f005:**
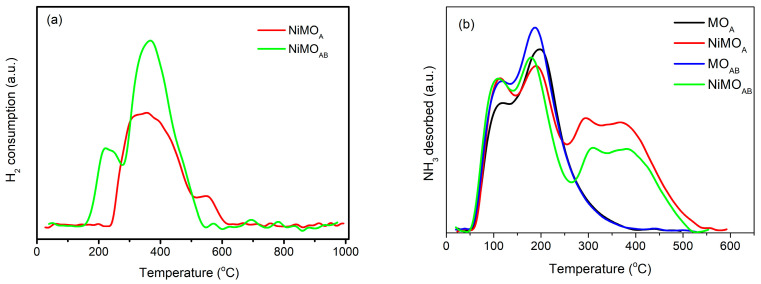
(**a**) H_2_-TPR and (**b**) NH_3_-TPD of activated mordenites and Ni-supported catalysts.

**Figure 6 nanomaterials-13-01603-f006:**
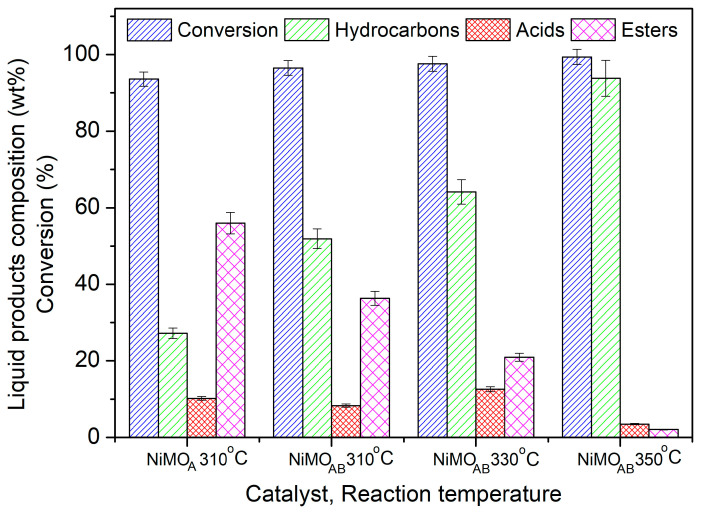
Composition of the liquid phase in hydrocarbons (n-alkanes), acids (palmitic and stearic acids) and esters (HMWE) as well as the conversion of biodiesel (fatty acid methyl esters) obtained over the NiMO_A_ and NiMO_AB_ catalysts. Reaction conditions: 100 mL biodiesel, 1 g catalyst; T: 310–350 °C; P_H2_: 40 bar; H_2_ flow 100 mL/min; reaction time: 9 h.

**Figure 7 nanomaterials-13-01603-f007:**
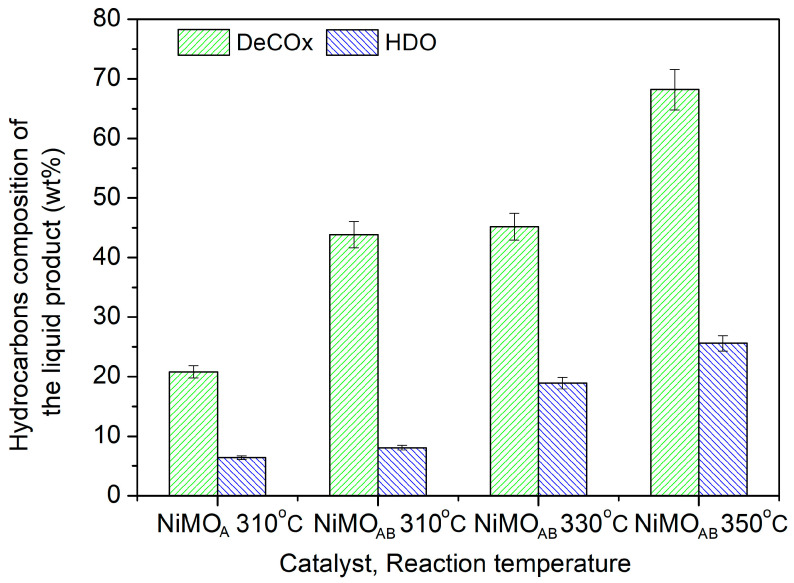
Contribution of the main SDO pathways (DeCOx, HDO) over the catalysts studied.

**Figure 8 nanomaterials-13-01603-f008:**
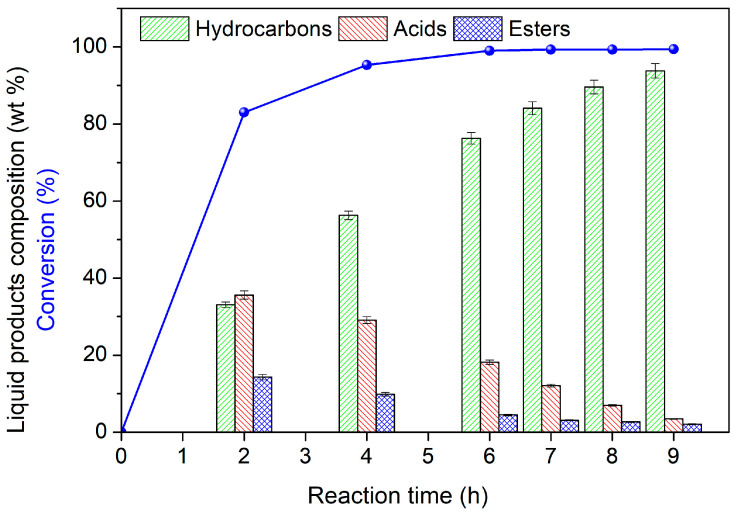
Composition of the liquid products in hydrocarbons, acids (mainly stearic and palmitic acid) and esters (HMWE) as well as the conversion of biodiesel obtained over the NiMO_AB_ catalyst. Reaction conditions: 100 mL biodiesel, 1 g catalyst; T: 350 °C; P_H2_: 40 bar; H_2_ flow rate: 100 mL/min.

**Figure 9 nanomaterials-13-01603-f009:**
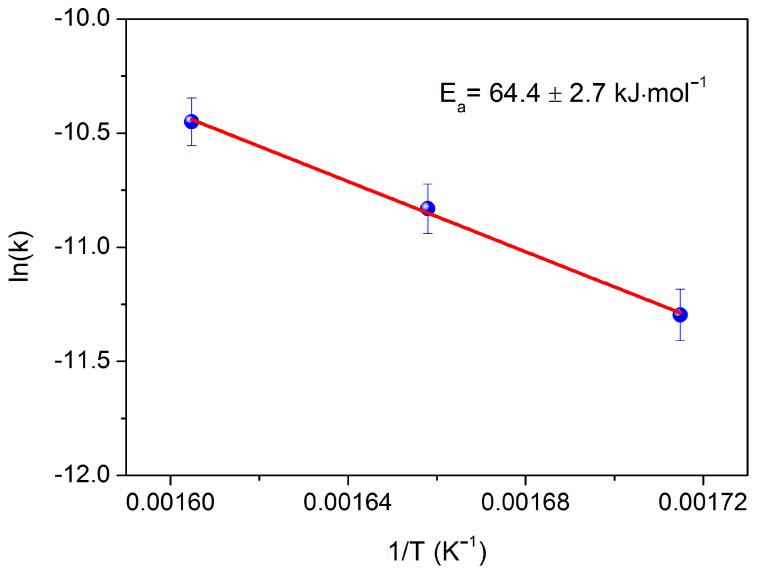
Arrhenius plot of apparent reaction constant (k) values. Reaction conditions: 100 mL biodiesel, 1 g catalyst; P_H2_: 40 bar; H_2_ flow rate: 100 mL/min.

**Table 1 nanomaterials-13-01603-t001:** Physicochemical characteristics of the catalysts. (Si/Al Ratio obtained using XRF analysis of the materials, total specific surface area, S_BET_; specific surface area of micropores, S_micro_; specific pore volume in meso- and macropores, V_BJH_; mean pore diameter of meso- and macropores, d_BJH_; mean crystal size of Ni^0^ nanocrystals determined using XRD, MCS_Ni_^0^ and Ni^0^ surface area, S_Ni_^0^).

Sample	Si/Al Ratio	S_BET_ (m^2^‧g^−1^)	S_micro_ (m^2^‧g^−1^)	^1^ S_Ni_^0^ (m^2^‧g^−1^)	V_BJH_ (cm^3^‧g^−1^)	d_BJH_ (nm)	MCS_Ni_^0^ (nm)
MO_nat._	4.59	16	8	-	0.06	19.5	-
MO_A_	5.85	156	131	-	0.09	20.4	-
MO_AB_	3.51	250	217	-	0.11	21.1	-
NiMO_A_	-	96	21	18.37	0.25	10.1	11
NiMO_AB_	-	124	73	20.21	0.20	14	10

^1^ S_Ni_^0^ was calculated considering spherical shape for Ni nanocrystals with radius equal to the half of MCS_Ni_^0^ value.

**Table 2 nanomaterials-13-01603-t002:** Acidic properties of activated mordenites and Ni supported on activated mordenite catalysts.

Sample	Total Acidity (a.u.)	Acid Site Concentration (%)			
		Physisorbed (104–108 °C)	Weak (183–196 °C)	Medium (292–309 °C)	Strong (367–437 °C)
**MO_A_**	385,090	-	94.2	5	0.8
	(498,440)	(22.7)	(72.8)	(3.9)	(0.6)
**MO_AB_**	414,151	-	92.5	7.3	0.2
	(527,211)	(21.4)	(72.7)	(5.7)	(0.2)
**NiMO_A_**	689,360	-	47.8	5.4	46.8
	(791,768)	(12.9)	(41.6)	(4.7)	(40.8)
**NiMO_AB_**	429,509	-	58.8	27.5	13.7
	(602,812)	(28.7)	(41.9)	(19.6)	(9.8)

**Table 3 nanomaterials-13-01603-t003:** Characteristics of the spent catalysts.

Sample	Reaction Temperature (°C)	S_BET_ (m^2^‧g^−1^)	S_Ni_^0^ (m^2^‧g^−1^)	^1^ C (wt.%)	^2^ d_Ni_^0^ (nm)
spNiMO_A_	310	29	11	9	19
spNiMO_AB_	310	22	14	7.8	15
spNiMO_AB_	330	17	8.5	6.6	24
spNiMO_AB_	350	16	8	4.6	26

^1^ Combustion elemental analysis; ^2^ metallic nickel mean particle size calculated using Scherrer’s equation.

## Data Availability

Not applicable.

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
