# Peer review of "Influence of Natural Mordenite Activation Mode on Its Efficiency as Support of Nickel Catalysts for Biodiesel Upgrading to Renewable Diesel"

_nanomaterials, 2023, doi:10.3390/nano13101603_

Round 1
Reviewer 1 Report
The manuscript «Influence of natural mordenite activation mode on its efficiency as support of nickel catalysts for biodiesel upgrading to renewable diesel» represents study of Ni-containing catalysts for biodiesel upgrading. Authors investigated two Ni-morderinite catalyst activated by acid and acid-alkaline pretreatment and showed that double activation enhances SSA, improves Ni reducibility, dispersion, resistance to sintering and low coke formation. Although, it is an interesting study, there are several aspects that should be reviewed before it can be accepted for publication.
3.1.3. Structural properties of the materials part. According to Figure 3, the positions of mordenite structure shifts after double activation. See, XRD patterns of MO-A and MO-B. Simultaneously, surface area drastically increases from 156 to 250 m2/g. Probably, some changes of mordenite structure occurs. Please comment these findings.
From XRD data, NiO and Ni phases are observed. Is it possible to estimate the ratio between phases.it would help to estimate the reduction degree.
From the text, it is not clear the reduced (reduced at 500°C for 2.5 h under H2) samples or after pretreatment in Ar are studied by TPR. In the case of reduced catalysts, the compassion the amount of NiO from XRD (see previous remark) and TPR would improve the clarity of manuscript.
3.3. Spent catalysts’ chareacteristics After catalytic tests significant catalyst sintering is observed. The surface area changed from 156-250 to ~20 m2/g. Simultaneously, the crystalline size of metallic nickel increases. However, according to XRD, the mordenite structure is stable. What is the origin of the sintering?
Reviewer 2 Report
In this submission (ID: nanomaterials-2346757), two kinds of nickel catalysts were prepared, and the catalysts were thoroughly characterized. Using the prepared catalyst, biodiesel yields could reach up to 94%. The catalyst system designed is superior to the traditional process and has broad prospects. The content of the article is good, but some parts need to be revised before publication.
1. The references on biodiesel catalysts in this paper suggest citing the latest research results (e.g., Bioresource Technology, 2023, 369, 128390; Progress in Energy and Combustion Science, 2016, 55, 98-194).
2. In line 30, please provide the basis for the instability of biofuels and quote references.
3. In lines 56-60, please explain in detail what microporous supports and mesoporous supports are and quote references.
4. In lines 212-213, why there is no absorption peak of NiMOA at 200 ℃?
5. In Figure 5 (b), why is the absorption peak resulting from the potent acid site at 400 ℃ smaller compared to that at 190 ℃?
6. In lines 279-280, “As hydrocarbons with odd carbon atoms are the main fraction of the hydrocarbons produced," what does this have to do with the contribution of the main SDO pathways?
7. In lines 326-327, "This phenomenon could be one of the reasons for the aforementioned decrease of the SBET" Please provide the basis for why you say so.
